# A Big Data-Based Commuting Carbon Emissions Accounting Method—A Case of Hangzhou

**Song Li** 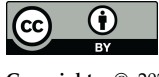**, Fei Xue, Chuyu Xia \*, Jian Zhang, Ao Bian, Yuexi Lang and Jun Zhou**

Faculty of Architecture, Civil and Transportation Engineering, Beijing University of Technology, Beijing 100124, China; lisong@emails.bjut.edu.cn (S.L.); xuefei@bjut.edu.cn (F.X.); zhjian@bjut.edu.cn (J.Z.); amhood1998@emails.bjut.edu.cn (A.B.); lyx@emails.bjut.edu.cn (Y.L.); zhouj@emails.bjut.edu.cn (J.Z.)
\* Correspondence: chuyu.xia@bjut.edu.cn

**Abstract:** Commuting carbon emissions are an essential component of urban carbon emissions, and determining how to reduce them is an area of great debate among researchers. The current research lacks a tool and instrument that can extensively account for residents' commuting. Traditional methods are mainly based on questionnaire surveys, which have low accuracy at spatial and temporal aspects. High accuracy carbon emission accounting methods can effectively assist urban planning and achieve precise urban emissions reductions. This study applies a taxi commuting carbon emissions accounting method divided into two main steps. Firstly, the carbon emissions of taxi trajectories are calculated using taxi trajectory data and a carbon emission calculation method developed based on VSP. Secondly, the taxi trajectory and POI data are used to filter the commuter trajectory with the help of a two-step moving search method. In this way, the taxi commuting carbon emissions were obtained. Then, the spatial distribution characteristics of residential taxi commuting carbon emissions are analysed by spatial autocorrelation tools, which could facilitate low carbon zoning management. A typical working day in Hangzhou was selected as the research object of this study. The results show that (1) morning peak commuting carbon emissions in the main urban area of Hangzhou reached 2065.14 kg per hour, accounting for 13.73% of all taxi travel carbon emissions; and evening peak commuting carbon emissions reached 732.2 kg per hour, accounting for 4% of all taxi travel carbon emissions; (2) At the grid level, the spatial distribution of commuting carbon emissions in Hangzhou shows a single central peak that decays in all directions; and (3) The results at the resident community scale show that urban public transport facilities influence resident community commuting carbon emissions. In areas such as at the urban-rural border, resident community commuting carbon emissions show high levels of aggregation, and in the main urban area, resident community commuting carbon emissions show low levels of aggregation. This study not only provides a new method of commuting investigation but also offers constructive suggestions for future carbon emission reduction under Hangzhou's urban planning.

**Keywords:** commuting carbon emissions; taxi GPS data; VSP model; spatial-temporal emissions distribution; Hangzhou

## 1. Introduction

The rapid growth of cities over the past 20 years has resulted in very significant carbon emissions [1–5]. Among various sectors, the transport sector accounts for 9% of total carbon emissions and has become a major contributor to China's environmental pollution [6–13]. Although commuting carbon emissions are not the largest share of transport carbon emissions, they have increased rapidly over the last 20 years. Furthermore, carbon emissions from transport and pm10 (10% of the total) are very harmful to humans [1]. At the same time, urban structures and carbon emissions of taxi commuting travel are closely related in terms of transportation carbon emissions, so it has become a major focus

of research to study the characteristics of carbon emissions from taxi commuting travel and to optimize urban structures to reduce carbon emissions.

Past studies on transport carbon emissions and residential commuting have relied heavily on research data [14]. These studies have suffered from poor data precision and are unrepresentative. In past estimates of transport carbon emissions, researchers have often estimated carbon emissions from large-scale data on overall fuel use and local research data; for example, vehicle and road statistics from relevant statistical yearbooks have been used to calculate vehicle transport carbon emissions [15], and some researchers have collected data from video road surveys lasting two weeks, as well as from parking data [16]. The data from these studies are not only unrepresentative on a large scale due to the small amount of data considered, but also have low precision due to the difficulty of obtaining technical information on vehicles and driving parameters as well as exhibiting low timeliness due to the time and effort required to collect data [17]. In studies on commuting, residential travel questionnaires and surveys have been widely used to investigate residential transport behaviour, with questionnaire data including commuting information, starting points, etc., but the data are very coarse, e.g., starting point information is only accurate to the street level, representing the smallest administrative unit in China [6]. For example, researchers at Peking University used data from the 2017 Resident Travel Survey and collected over 500 questionnaires from different groups. Differences in commuting distances, times, and modes of transport between different income classes and different distributions of people were studied [6]. Wang et al. investigated the characteristics of urban commuting and the factors that influence them through a resident questionnaire [18]. Moreover, ref. [19] used the data from the National Household Travel Survey in 2017 to find the relationship between commuters' backgrounds and the external built environment through structural modelling. These studies suffer from a lack of representativeness on a large scale due to the use of small samples, and they also suffer from time-consuming and costly data collection methods, inaccurate data, and poor timeliness.

With the prevalence of information technology, an increasing amount of big data is being used by researchers, among which taxi big data are of great interest [20,21]. First, taxi data contain high-precision trajectory points of passengers' entire journeys, such as data on instantaneous speed, mileage, direction, and other important information, allowing for accurate carbon emission estimation [21]. Second, taxis are closely related to residents' lives. Numerous studies have found a strong daily routine regardless of the location and frequency of boarding and arrival [22–27]. Third, taxi data are controlled by transport sector companies, which are private, and the possession of such data by the transport sector has a direct effect on reducing emissions. Some studies have relied on taxi GPS data to estimate the spatial and temporal distribution of taxi carbon emissions and their characteristics in Shanghai [8]. Finally, taxi data also have the potential to be used to make trip-type judgements, as taxi estimation data include pick-up/drop-off point data. For example, types of taxi trips have been determined by the distributions and types of taxi drop-off points and surrounding POIs [27,28]. Due to the many advantages of taxi GPS data for carbon emission studies, we propose using these data for subsequent studies.

Current research supports the use of taxi GPS data to analyse urban travel patterns and cab travelling's carbon emissions. However, there are limitations to the current related research. This study uses big data to mine the characteristics of carbon emissions from the taxi commuting travel of Hangzhou residents. A framework for estimating taxi commuting carbon emissions using multiple big data sources, such as POI and taxi GPS data with G2SFCA and VSP models is first proposed. In this study, the GPS trajectory data of 1752 taxis in the city of Hangzhou on a typical weekday in 2017 were extracted for trajectory classification and trajectory carbon emission estimation, and commuting carbon emissions were aggregated to resident communities. Finally, the ArcGIS 10.6 platform and Geoda 1.18 software were used to analyse geographical and spatiotemporal characteristics and, finally, the characteristics of Hangzhou taxi commuting carbon emissions were derived.

The remainder of the article is arranged as follows. Section 2 shows the big data and methods that this research used; data used include taxi trajectory and POI data and methods used include the VSP model-based motor vehicle carbon emission accounting method, the estimation method of taxi trajectory classification based on G2SFCA, Geoda1.18-based spatial autocorrelation analysis, and ArcGIS 10.6-based carbon emission model mapping. Sections 3–5 present the study's experimental results and related discussion and conclusions.

## 2. Materials and Methods

### 2.1. Framework

The research process of the article is shown in Figure 1, which is divided into three main parts, the calculation of taxi carbon emissions, the identification of taxi commuting trajectories, and the spatial analysis of taxi commuting carbon emissions. The first is to calculate the carbon emissions of each trajectory based on the taxi trajectory data using a VSP-based carbon emission model. This method will be described in Section 2.4.1, and the results will be presented in Section 3.1.1. The G2SFCA (two-step floating catchment area method) will then be then used to identify commuter taxi tracks by using the pick-up and drop-off points of the taxi tracks and the POI data. This part of the methodology will be described in Section 2.4.2, and the results will be presented in Section 3.1.2. Finally, we aggregate taxi commuting carbon emissions to residential communities. We use a spatial autocorrelation tool to analyze the spatial characteristics of commuter carbon emissions. The methodology used in this section will be described in detail in Section 2.4.3. The results will be presented in Section 3.2.

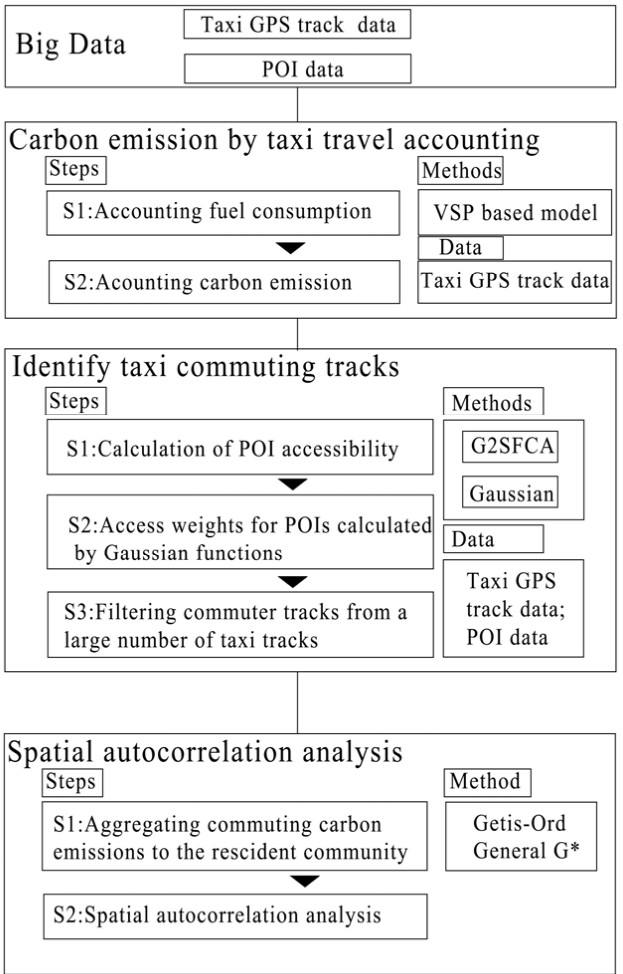

**Figure 1.** Research Process.

## 2.2. Study Area

The chosen study area is Hangzhou, Zhejiang Province, China, which, as the capital of Zhejiang Province, is one of the most prosperous cities in China, with an area of 16,596 km$^2$ and a population of 7,539,000 (2017). We selected the main urban area of Hangzhou as the study area in this research, The residents are mainly concentrated in the city. Hangzhou is one of the first low-carbon pilot cities identified by the National Development and Reform Commission, so studying commuting carbon emissions in this region is essential. The location and urban structure of Hangzhou are shown in Figure 2.

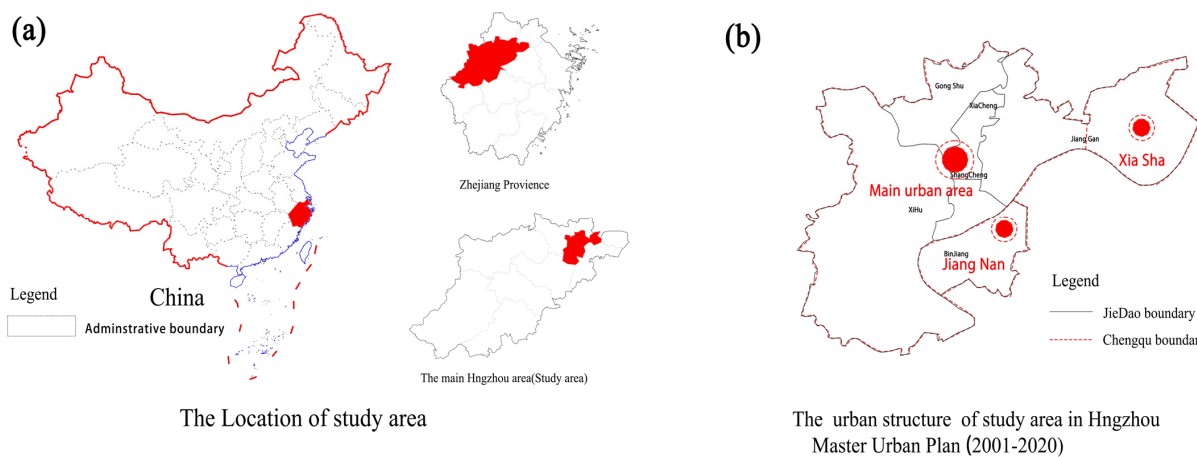

**Figure 2.** Location of the study area and Hangzhou Master Urban Plan of 2001–2020. ((**a**) shows the location of the study area in China, Zhejiang and Hangzhou respectively; (**b**) shows the structure of the region, the study area consists of three main centres, which are described in detail below).

According to the spatial structure of the study area under the Hangzhou Urban Master Plan, the spatial structure of Hangzhou includes one central area and three secondary areas, of which the study area mainly contains the "one main area" in the spatial structure plan, i.e., the central city and the northern areas of Xiasha Cheng and Jiangnan Cheng, which are approximately 15 km away from the main urban area.

The central city includes the West Lake District, Gongshu District, and Shangcheng District. The central city mainly contains the functions such as residential and administration offices; commerce and finance; science and technology; education, culture, and urban high-tech industries, forming the core of Hangzhou. Xiasha Cheng is a new city dominated by the Hangzhou Economic and Technological Development Zone and college and university campuses. Jiangnan Cheng is a modern city focused on science and technology and a distant commercial centre with high-tech industrial parks as its major backbone, with urban industry, teaching, and research and development occurring in coordination with each other. The study area includes the northern area of Jiangnan Cheng; the Binjiang District, which functions mainly as a high-tech industrial park hosting high-tech firms such as such as Alibaba; and a small number of residential communities.

## 2.3. Data Collection

### 2.3.1. Taxi GPS Data

The taxi GPS data for this study was acquired from the Comprehensive Information Centre of the Hangzhou Municipal Bureau of Transportation. The GPS trajectory data of taxi trips occurring from 6:30 a.m.–12:00 a.m. on a typical working day in 2017 were selected. The taxi trajectory GPS data were transmitted from each vehicle's GPS to the server, including real-time location (latitude and longitude), instantaneous speed, vehicle status, license plate number, vehicle status information, time, and other taxi data. The time interval between each upload of GPS information by a taxi ranged between 30 and 40 s.

This study used data for 6:30 a.m.–12:00 a.m., as shown in Table 1, amounting to 1,111,291 entries.

**Table 1.** Original taxi trajectory information (*** is a hidden taxi license plate).

| Point Number | Licence Plate | Type | Longitude | Latitude | Speed (km/h) | Direction | State | ROAD_ ID | GPS_TIME | UPDATE_ TIME |
|---|---|---|---|---|---|---|---|---|---|---|
| 1370 | AUT *** | 1 | 120.195694 | 30.35249 | 55 | 246 | 1 | −1 | 05-6-2017 13:36:42 | 05-6-2017 13:36:47 |
| 1887 | AUT *** | 1 | 120.18566 | 30.306992 | 52 | 268 | 1 | 7550 | 05-6-2017 10:22:00 | 05-6-2017 10:22:02 |
| 1903 | AUT *** | 1 | 120.20334 | 30.308031 | 0 | 276 | 1 | 7479 | 05-6-2017 10:17:00 | 05-6-2017 10:17:03 |
| 2095 | AUT *** | 1 | 120.23065 | 30.36269 | 53 | 290 | 1 | 795 | 05-6-2017 15:02:22 | 05-6-2017 15:02:24 |

To study residents' commuting trips, taxi travel data for the hours of 6:30–10:00 and 16:30–20:00 were screened, and intermittent trajectory information was processed into coherent taxi trajectory information using the TRACK tool and PYTHON in the GIS platform. The trajectories that were not operational and lasted less than 2.5 min were then excluded. A total of 25,089 trajectories (6:30–10:00 and 16:30–20:00) were obtained in this study, and the time information of the trajectory points was used to recognize the pick-up and drop-off points, as shown in Table 2.

**Table 2.** Taxi trajectory data.

| Trajectory ID | Start Time | End Time | Pick up Point Number | Drop off Point Number | Carbon Emissions (kg) | Distance (km) |
|---|---|---|---|---|---|---|
| 34 | 05-6-2017 17:49:26 | 05-6-2017 17:56:07 | 34,677 | 63,090 | 0.551884 | 1.553232 |
| 46 | 05-6-2017 16:29:48 | 05-6-2017 16:34:08 | 7324 | 83,889 | 0.412152 | 1.748736 |
| 57 | 05-6-2017 17:30:25 | 05-6-2017 17:47:08 | 101,532 | 78,115 | 1.365418 | 3.655797 |

2.3.2. POI Data

The POI data that we used comes from the Gao-De platform, crawled through the API interface of the Gaode developer platform, and include the POIs for the whole area of Hangzhou, covering a total of 20 major categories (first-level classification). We use POIs with the two-step floating catchment method to classify taxi trajectories (as explicitly described in the Section 2). Therefore, the initial screening and classification of Hangzhou POIs according to their functions and opening times [27] and other specifications are detailed in Section 2.4.2 below.

*2.4. Method*

2.4.1. Taxi Carbon Emission Modelling

Among the models for estimating transport carbon emissions, carbon emission models based on the VSP model mainly applied to estimate the relationship between vehicle operating conditions and fuel consumption, are becoming increasingly popular among researchers. On the one hand, the VSP model reflects the relationships between specific power and acceleration, speed, and gradients. On the other hand, the VSP model also strongly correlates with fuel exhaustion, so this study develops a fuel consumption model for taxis based on the VSP model.

According to [29], the VSP model for light vehicles is simplified and expressed in the following equation.

$$VSP = v \cdot [1.1a + 0.98a \cdot grade(\%) + 0.132] + 0.000302 \cdot v^3 \tag{1}$$

where $v$ is the speed of the motor vehicle in m/s. $a$ is the transient acceleration of the motor vehicle in m/s$^2$. *grade* is the gradient (dimensionless parameter), which should be taken as 0.0 in the application.

We used an improved VSP-based model to develop a vehicle carbon emission estimation model [28]. There are two main steps. First, the fuel consumption of vehicle driving is derived

from speed and mileage information from GPS data. Since VSP strongly correlates with fuel consumption, it is difficult to acquire the vehicle operating conditions required for VSP in this study. Therefore the fuel consumption rates in this study use the results of Zhao t et al. as shown in Table 3 [30]. Second, the corresponding carbon emissions are calculated from fuel consumption.

**Table 3.** Time constraints of POI points.

| Types | Time Periods | Pick Up/Drop off Points | POI Types | POI Names |
|---|---|---|---|---|
| From home to work | 6:30 a.m.–10:00 a.m. | Pick-up points | Residence | Commercial Residence |
| | | Drop-off points | Work | Catering Services |
| | | | | Road appurtenances |
| | | | | Public Facilities |
| | | | | Enterprises |
| | | | | Shopping facilities |
| | | | | Financial and Insurance Services |
| | | | | Scientific, educational and cultural services |
| | | | | Motorbike services |
| | | | | Car Services |
| | | | | Car repairs |
| | | | | Automotive sales |
| | | | | Live Services |
| | | | | Sports and leisure services |
| | | | | Government Agencies and Societies |
| | | | | Accommodation services |
| | | | Medical | Health Care Services |
| | | | Transport | Transport Facility Services |
| From work to home | 4:30 p.m.–8:00 p.m. | Pick-up points | Work | Catering Services |
| | | | | Road appurtenances |
| | | | | Public Facilities |
| | | | | Enterprises |
| | | | | Shopping facilities |
| | | | | Financial and Insurance Services |
| | | | | Scientific, educational and cultural services |
| | | | | Motorbike services |
| | | | | Car Service |
| | | | | Car repairs |
| | | | | Automotive sales |
| | | | | Life Services |
| | | | | Sports and leisure services |
| | | | | Government Agencies and Societies |
| | | | | Accommodation services |
| | | | Medical | Health Care Services |
| | | | Transport | Transport Facility Services |
| | | Drop-off points | Restaurant | Catering Services |
| | | | Shopping | Shopping facilities |
| | | | Transport | Accommodation services |
| | | | Education | Scientific, educational and cultural services |
| | | | Residence | Commercial Residence |
| | | | Medical | Health Care Services |
| | | | Entertainment | Scenic areas |
| | | | | Sports and leisure services |
| | | | | Life Services |

First step: Calculating fuel consumption for taxi journeys [31].

$$Q = ER_0 \cdot NER_I \cdot TI \tag{2}$$

where $Q$ is the actual fuel exhaustion of the taxi in grams. $ER_0$ is the fuel exhaustion rate when $VSP$ equals zero using parameter 0.274 g/s. $NER_i$ is the standard fuel exhaustion rate in Table 4 when the speed is interval $i$ and reflects dimensionless data. $T_I$ is the total length of time, in seconds, that the taxi actually travels at an average speed in interval $i$.

**Table 4.** Standard fuel exhaustion rate.

| i | Average Speed Interval i (km/h) | NER | i | Average Speed Interval i (km/h) | NER |
|---|---|---|---|---|---|
| 1 | 0–2 | 1.085 | 20 | 38–40 | 2.395 |
| 2 | 2–4 | 1.258 | 21 | 40–42 | 2.441 |
| 3 | 4–6 | 1.311 | 22 | 42–44 | 2.47 |
| 4 | 6–8 | 1.477 | 23 | 44–46 | 2.538 |
| 5 | 8–10 | 1.573 | 24 | 46–48 | 2.566 |
| 6 | 10–12 | 1.645 | 25 | 48–50 | 2.581 |
| 7 | 12–14 | 1.729 | 26 | 50–52 | 2.595 |
| 8 | 14–16 | 1.807 | 27 | 52–54 | 2.68 |
| 9 | 16–18 | 1.841 | 28 | 54–56 | 2.716 |
| 10 | 18–20 | 1.923 | 29 | 56–58 | 2.756 |
| 11 | 20–22 | 1.997 | 30 | 58–60 | 2.81 |
| 12 | 22–24 | 2.045 | 31 | 60–62 | 2.864 |
| 13 | 24–26 | 2.092 | 32 | 62–66 | 2.956 |
| 14 | 26–28 | 2.163 | 33 | 66–70 | 3.05 |
| 15 | 28–30 | 2.187 | 34 | 70–80 | 3.289 |
| 16 | 30–32 | 2.251 | 35 | 80– | 3.551 |
| 17 | 32–34 | 2.329 | | | |
| 18 | 34–36 | 2.338 | | | |
| 19 | 36–38 | 2.391 | | | |

Second step: Calculation of taxi carbon emissions based on fuel consumption.

$$E = Q \cdot EF_K \tag{3}$$

$E$ denotes the carbon emissions generated per trajectory, and $EF_k$ is the emission factor for emissions, which is 2.98 kg/L for $CO_2$ and a conversion factor of 0.27 between $CO_2$ and carbon, according to Xia et al. in their 2017 study [28,31].

### 2.4.2. Identification of Travel Origins and Destinations

As the time characteristics of commuting behaviour are more pronounced, this study defines two modes of commuting behaviour by referring to Zhao et al.'s time constraint for POIs [27]. They include travel from home to work, which occurs between 6:30–10:00, and travel from work to home, which occurs between 16:30–20:00. To filter out trajectories generated by commuting among all taxi trajectories, this study uses the time and space-based and POI point-of-interest travel inference method proposed by [27]. The method used in this study calculates the probability of visits for each POI within walking distance (500 m) of the drop-off point through the G2SFCA method with a Gaussian decay function. Some modifications are made to Zhao et al.'s method, which is also divided into three steps: identifying candidate POIs, calculating POI accessibility, and calculating POI visitation weights.

In identifying candidate POIs, mainly considering the more mixed distribution of POI points in urban space, this study introduces temporal constraints to filter POIs [27,28]. The rationale is that different functions provide specific services at different times. For example, in the case of restaurants and shopping centres, although the two are very similar in terms of space taken, restaurants only offer food and beverage services at midday.

Shopping centres are open for longer hours (10:00–21:00). Therefore, at specific times of the day, e.g., in the morning, restaurant POIs are set as noncandidate POIs, while shopping POIs are identified as candidate POIs. The classification by Zhao et al. is limited to determining drop-off points. In this study, destination POIs are extended to departure point and destination POIs with separate time constraints based on commutes. See Table 3 for details.

For the measurement of the access weight of a POI, there are three steps: first, for each pick-up or drop-off points j, search for POI points i within a certain radius and calculate supply/demand ratio $R_j$. Second, for each POI point $i$, search for pick-up or drop-off point $j$ within walking radius and sum supply/demand ratios $R_j$ to obtain the accessibility $A_i$ of the POI. Finally, access weight $W_{ij}$ of drop-off/boarding point j for the POI points within the search radius is calculated with a Gaussian function, where the enormous access weight is the type of destination/starting point. The following four main steps identify the trip type.

First step: Calculate POI point supply/demand ratio $R_j$ within search threshold $d_0$ for single drop-off/pick-up point $j$.

$$R_\mathrm{j} = \frac{1}{\sum\limits_{i \in \{d_{ij} \leq d_0\}} G(d_{ij}, d_0) \times 1} \tag{4}$$

Second step: Calculate supply/demand ratio $A_i$ for each POI facility point $i$.

$$A_\mathrm{i} = \sum_{j \in \{d_{ij} \leq d_0\}} G(d_{ij}, d_0) \times R_j \tag{5}$$

where $R_j$ is the supply/demand ratio of the boarding/disembarking points for the POI points in the range. Equation (6) is a commonly used Gaussian distance decay function and is widely used with the 2SFCA method [32–34].

$$G(d_{ij}, d_0) = \begin{cases} \dfrac{e^{-\frac{1}{2} \times (\frac{d_{ij}}{d_0})^2} - e^{-\frac{1}{2}}}{1 - e^{-\frac{1}{2}}}, & d_{ij} \leq d_0 \\ 0, & d_{ij} > d_0 \end{cases} \tag{6}$$

Third step: Calculate the visit weight of each pick-up/drop-off point j for facility point $i$ within search radius $d_0$ and determine types of start- and endpoints.

$$W_\mathrm{ij} = G(d_{ij}, d_0) \times A_i \tag{7}$$

For POI point $i$ within the search range of pick-up or drop-off points $j$, the attenuation of the reachability $Ai$ of the POI on $d_{ij}$ is calculated. The value determined after the attenuation calculation is noted as access weigh t $W_{ij}$ for visit i of j (Equation (7)). The attenuation is done using the Gaussian attenuation function mentioned above, which has a normal distribution of the computed results and is widely used in various calculations regarding distance attenuation [32–34].

For the POI points within the search range of pick-up/drop-off points, the POI point in which *j* has the largest visit weight to *i* is selected as the type of pick-up/drop-off point.

Final step: The type of track is determined by the type of POI to which the start- and endpoints belong: From home to work and from work to home.

2.4.3. Spatial Autocorrelation Analysis of Carbon Emissions from Residential Communities

This study uses the ArcGIS 10.6 platform to visualize the carbon emissions of taxi commuting, aggregating the carbon emissions of each small segment of the trajectory to the 1 km × 1 km grid to which it belongs.

Subsequently, the spatial distribution characteristics of carbon emission by taxi commuting travel in residential communities were explored. The carbon emissions of the total taxi trajectory were aggregated to the residential community. The spatial autocorrelation software GeoDa1.18 was used to conduct a Getis-Ord Gi analysis of residential taxi commuting carbon emissions to obtain the spatial aggregation characteristics of residential commuting carbon emissions.

The spatial autocorrelation mainly involves the Getis-Ord General Gi* assessment with the following correlation Equation (8):

$$Gi^* = \frac{\sum\limits_{j=1}^{N} W_{ij} \cdot X_j - \overline{X} \sum\limits_{j=1}^{N} W_{ij}}{\sqrt{\frac{\sum\limits_{j=1}^{N} X_j^2}{N} - (\overline{X})^2} \cdot \sqrt{\frac{N \sum\limits_{j=1}^{N} W_{ij}^2 - (\sum\limits_{j=1}^{N} W_{ij})}{N-1}}}, \forall j \neq i \tag{8}$$

In Equation (8), N is the number of residential points. $X_i$ and $X_j$ are the carbon emissions of residential points *i* and *j*, respectively. $\overline{X}$ is the average of carbon emissions. $W_{ij}$ is the spatial connection matrix between residential points *i* and *j*.

## 3. Results

### 3.1. Accounting for Residential Commuting Carbon Emissions of Taxis

3.1.1. Carbon Emissions of Taxi Travel

In this section, we apply the taxi carbon emission model mentioned in Section 2.4.1 to estimate the total carbon emissions by a cab's trajectory on a typically weekday in 2017, The results are shown in Figure 3 below.

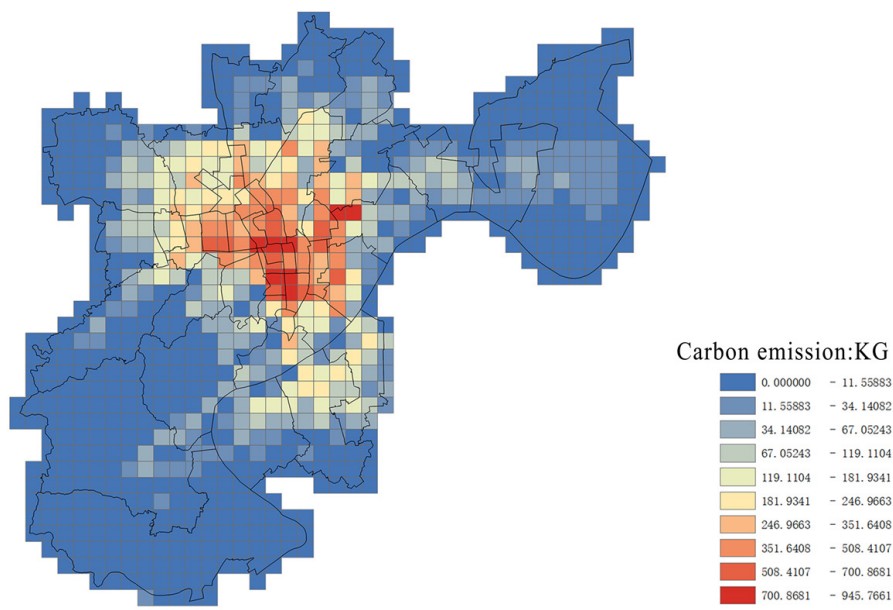

6:30-24:00 Spatial dispersed of carbon emissions by taxi

**Figure 3.** Spatial dispersion of all taxi carbon emissions.

The carbon emissions of taxis for the whole day from 6:30 a.m.–12:00 a.m. are 52,934 kg. From Figure 3, it can be observed that the carbon emissions of taxis are mainly concentrated near the main urban area centred on West Lake, and the total carbon emission trend belongs to the high centre and gradually spreads to the periphery. The subsequent higher carbon

emissions are found in the Binjiang district on the north side of Jiangnan Cheng, while carbon emissions in Xiasha Cheng are lower.

In terms the spatial and temporal distributions of taxi emissions in the three cities, the main urban area and Jiangnan Cheng have higher carbon emissions. In contrast, Xiasha Cheng, a 'secondary city', has lower carbon emissions than Jiangnan Cheng, also a 'secondary city'. In terms of the functions of the main urban area, Xiasha Cheng and Jiangnan Cheng, and the spatial distance between them, the main urban area is positioned for technology production, business, finance, and living and residential functions. Xiasha Cheng is positioned as a high-tech industrial park and high education park.

### 3.1.2. Estimated Types of Residential Trips

In this section, the two-step mobile search method is applied to classify the taxi trip trajectories of a typical weekday in 2017, and the results are shown in Table 5.

**Table 5.** Results of Taxi Commute Carbon Emissions.

| Duration | Type | Quantity | Ration (Count) | Carbon Emissions (kg) |
|---|---|---|---|---|
| | Commuting (From home to work) | 8738 | 78.60% | 7228 |
| 6:30–10:00 | Others | 2320 | 20.89% | 1929 |
| | Total | 11,103 | 100% | 9157 |
| | Commuting (From work to home) | 2610 | 17.74% | 2563 |
| 16:30–20:00 | Others | 12,096 | 82.25% | 11,694 |
| | Total | 14,706 | 100% | 14,257 |

Table 5 shows that taxi commuting accounts for approximately 78% of all taxi travel from 6:30 a.m.–10:00 a.m. From 4:0 p.m.–8:00 p.m., taxi commute trips account for only 17.74% of all taxi travel. The number of trajectories from 6:30 a.m.–10:00 a.m. is similar to the number of trajectories collected for 4:30 p.m.–8:00 p.m. Thus, most taxi commuting is done in the morning from 6:30 a.m.–10:00 a.m., i.e., from home to work, while from 4:30 p.m.–8:00 p.m., when travel from work to home occurs, commuting is no longer the primary purpose of taxi travel.

### 3.1.3. Estimated Residential Commuting Carbon Emissions of Taxi Travel

From the results in Sections 3.1.1 and 3.1.2, we obtain residents' taxi commute carbon emissions. The spatial autocorrelation of the residents' taxi commuting carbon emission are shown in Figure 4 below.

For commuting purposes, total carbon emissions accounted for 18.6% of total carbon emissions (without screening short trajectories) at 9157 kg. The total commuting carbon emissions in the morning peak were 7226 kg, averaging 2064.57 kg per hour; the total carbon emissions for commuting in the evening peak were 2562 kg, averaging 732 kg per hour. As shown in Figure 4e,f, during the morning peak period, the number of taxis for commuting purposes accounted for 78% of the morning peak taxi trajectory, when most taxi trips were of the commuting type. As a result, as shown in Figure 4a,c, the spatial dispersal of taxi's commuting carbon emission from 6:30 a.m.–10:00 a.m. is mainly concentrated in the main urban area and the north side of Jiangnan Cheng, where the peak carbon emissions reach an average of over 100 kg in the grids. As shown in Figure 4b,d, from 4:30 p.m.–8:00 p.m., the taxi trajectory for commuting purposes accounted for 17.74% of the total taxi trajectory in the evening peak. The main purpose of taxi trips is not commuting, and the spatial distribution of carbon emissions in this period is still concentrated in the main urban area, with sporadic distribution in XiaSha Cheng.

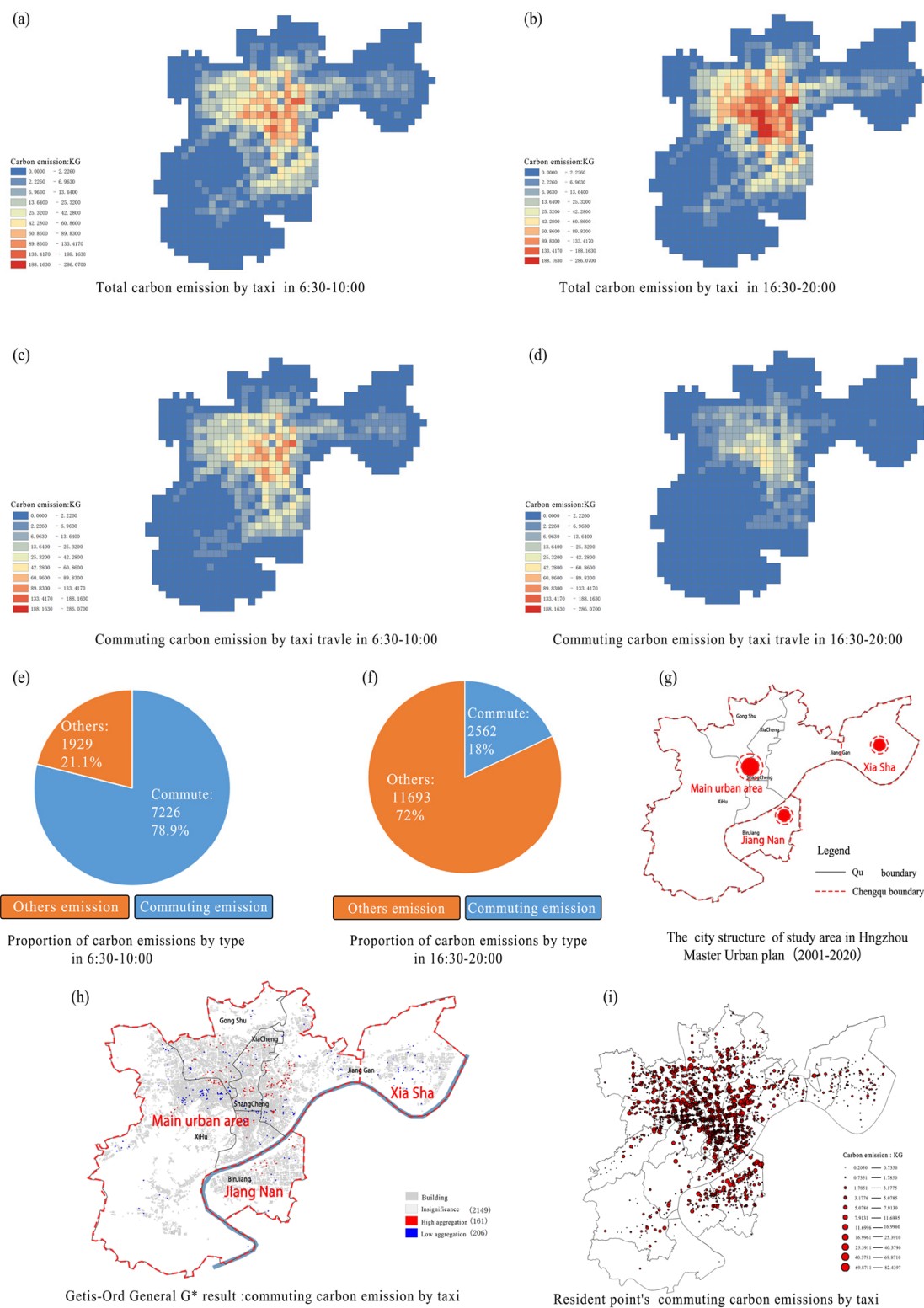

**Figure 4.** Spatial distribution of carbon emission by different type of taxis travel on a typical work day and its Getis—Ord General G* result. ((**a**–**d**) shows the spatial distribution of carbon emission in different types, (**e**,**f**) shows the proportion of carbon emission by type. (**g**) is the city structure oof study area. (**h**,**i**) shows the resdient communities carbon emission and it's result of Spatial autocorrelation).

*3.2. Spatial Autocorrelation Analysis of Carbon Emissions from Residential Point Taxi Commuting*

In this section, taxi commuting carbon emissions are aggregated to residential points. A spatial autocorrelation analysis explores the relationship between the spatial aggregation and spatial distribution of taxi commuting carbon emissions at residential communities. The spatial autocorrelation results are shown in Figure 4h,i.

In Figure 4, (i) is the sum of commuting carbon emissions by taxi travel from 6:30 a.m.–10:00 a.m. and 4:30 p.m.–8:00 p.m. on a typical weekday and is the result of a hierarchical symbolic visualization derived from ArcGIS, where the more carbon emissions there are at a round point, the larger its rounded area is; (h) shows the result of the Getis-Ord General Gi* of point (i).

Two results from this subsection are worth noting. First, the results of the distribution of the high and low clustering of carbon emissions from residential commuting (Figure 4h), the low and low clustering of carbon emissions is mainly concentrated within the main urban area, with a small number of clusters located in areas far from the main urban area.

Second, the majority of residential communities with high and high carbon emissions are concentrated within the urban-rural interface of the main urban area. For example, in the western side of the Jianggan District, in the urban-rural area near agricultural land, as shown in Figure 4h, buildings are sparse. Some residential points in Xiasha Cheng, approximately 15 km away from the main urban area, show a clear low-low agglomeration pattern. On the other hand, Jiangnan Cheng, a subcity, is only separated from the main urban area by the geographical factor of the Qiantang River, but its residential community distribution shows distinctly high clustering.

## 4. Discussion

*4.1. Discussion of the Results of the Proposed Carbon Emissions Accounting for Taxi Trips Based on the VSP Model*

4.1.1. Comparison of Traditional Carbon Emission and VSP Model-Based Carbon Emission Estimations

The method proposed offers the advantages of high research accuracy, large-scale analysis, broad representation, and real-time performance relative to the use of traditional research data such as yearbook and fuel consumption data for carbon emission estimation.

The data used for carbon emission estimation are the big data of taxi trajectories from taxi companies. Based on the VSP model, carbon emissions are calculated from acceleration, speed, mileage, time, and vehicle driving condition data from GPS information to calculate fuel consumption data for each small section of a taxi trajectory. Due to the high refresh rate (20–40 s) and accuracy of GPS positioning, the data volume is large and representative of a broad range.

Using traditional methods of calculating motor vehicle carbon emissions, Ref. [35] accounted for transportation carbon emissions in Florida in 2011. Using data on population, road miles, numbers of cars and trucks, and fuel economy for 67 counties, the VMT model was applied to estimate carbon emissions in Florida [35]. However, this method's lowest resolution of transport carbon emission results is only of the county scale. Data are also available on an annual basis and are time-lagged. Compared to the VSP model-based method proposed in this study, this method can accurately account for carbon emissions to a hundred-meter level and is more time sensitive than the previous method. The same problem affects other studies, such as those based on vehicle and road statistics from relevant statistical yearbooks [15], video road survey data, and parking data [16]. Therefore, the present method has the advantages of achieving high levels of research accuracy, a large scale, wide representation, and real-time performance.

4.1.2. Comparison of Taxi and Traditional Commuting Investigations

This study uses a two-step floating catchment method and taxi track data to screen a large number of taxi commuters. Compared with traditional commuting research, this

method has the advantages of using a wide range of samples, obtaining process data, using accurate data, and obtaining data in a timely manner. In particular, by combining the identification of commuting trajectories with the accounting method of commuting carbon emissions by taxi travel, commuting carbon emissions can be accurately obtained.

There are few studies on commuting and commuting carbon emissions. Previous studies mainly use residents' travel questionnaires and other tools to obtain information on residents' travel destinations, starting places, and transportation modes. However, the accuracy of starting places and destinations is usually poor and usually accurate to the street level [6], and critical data such as traffic trajectories are lost. Wang et al. studied the characteristics of resident commuting research in China using questionnaires in 2018 [18]. Lou et al. studied the commuting patterns of different income groups, and their new town research examines the commuting characteristics of people of different income groups and from different types of new towns [6]. These studies accurately portray the effects of different commuting groups' social and economic characteristics on commuting [19]. However, they do not accurately account for the carbon emissions of commuting, and they are inadequate in terms of sample size, the precision of commuting and carbon emissions of the commuting process.

This study uses a two-step floating search method to calculate types of taxi trajectories by using the POIs of the pick-up/drop-off points of taxi trajectories and surrounding areas. It is possible to filter out a large volume of commuting samples from a large number of taxi tracks. On the one hand, this makes up for the lack of information about the commuting process in questionnaire research, and on the other hand, it avoids the problems of small sample sizes and the loss of commuting information in traditional research to obtain commuting samples. Combined with the VSP model based on the taxi carbon emission accounting method mentioned in the previous section, the approach can accurately obtain the carbon emissions of residents' taxi commuting.

### 4.2. The Impact of Urban Spatial Structure and the Construction of Urban Public Transport Facilities on Carbon Emissions by Taxis

This study finds that the low clustering of commuting carbon emissions in the main urban area and high clustering in the main urban area periphery may be due to a lack of public transport in the urban periphery (Figure 5). The results also show that urban structures may impact commuting carbon emissions. Future urban planning measures should increase the construction of public transport facilities at the urban-rural interface and, at the same time, focus on balancing jobs and housing to promote a rational mix of functions.

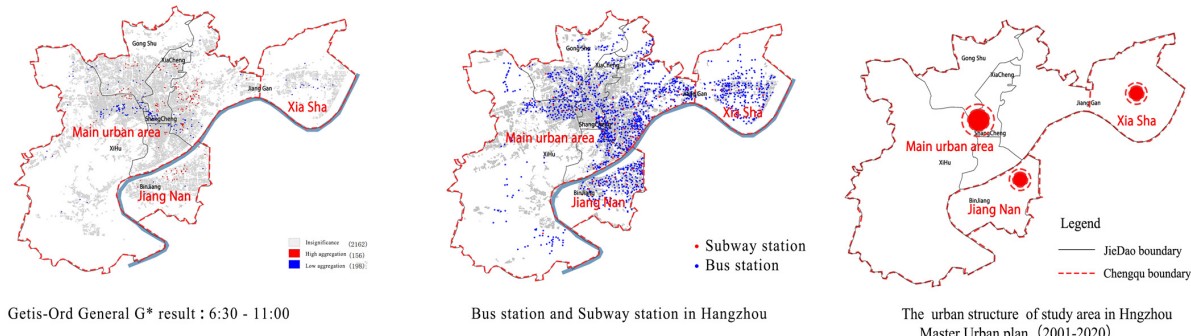

**Figure 5.** Comparison of the spatial distributions of residential points' commuting carbon emissions and bus stops.

According to (i,h) in Figure 4, the high clustering of commuting carbon emissions from residential communities mainly occurs in the urban-rural border areas in the main urban area, such as on the western side of the main urban area and in the Jianggan District, which are urban-rural border areas with minor public infrastructure construction located farther

from the main urban area. The main function is residential communities, and low-low aggregation of carbon emissions from commuting to residential locations mainly occurs in the core areas of the central city, such as West Lake and the south side of the Shangcheng District. These areas are mainly business cores with many residential locations. This may be caused by the insufficient number of public transport facilities in suburban and urban-rural areas, with the shortfall being filled by other forms of transport. The phenomenon is relevant to studies of areas such as Chengdu and Seoul. After Li et al. identified GWR models for online taxi trips and critical factors in Chengdu, they found that online taxi suburban public transport complements other services [36]. Nam et al. applied taxi GPS data signals in Seoul, South Korea to analyse taxi passengers' origins and spatiotemporal variation. Their results indicate that traditional taxis form a complementary relationship with underground trips in places where subways are less common [37]. Other studies on taxis conclude that net cars and taxis complement bus metro lines. When there is a shortage of bus metros, the ridership density of net cars and taxis is higher [38–40]. It may be that public transport systems do not have an advantage in terms of long-distance commuting, with numerous studies showing that residents prefer to take taxis over buses for long-distance commuting [41] and that high-income residents of new city areas prefer other types of travel such as car and taxi transport [6,42], as this not only eliminates the need for intermediate transfers but also makes travel more comfortable overall. When we compare Figure 5, the aggregated relationship between rail transit distribution and the commuting pollutions of carbon by residential taxi transports, we also find that residential commuting carbon emissions by taxi travel are lower in areas with a higher density of public transport facilities. This phenomenon is consistent with previous studies. Therefore, higher carbon emissions from taxi commuting in urban fringe settlements shown in Section 3.2 are, on the one hand, due to the inadequate construction of public infrastructure in the urban fringe and, on the other hand, likely due to the overall public transport system experience not being as optimised as that of taxis. Therefore, in the future, in Hangzhou's master urban planning, more transport facilities should be built within the urban-rural boundaries of the central city area, including on the northern side of Shangcheng District and the west side of the Jianggan District. At the same time, the operating standards of public transport operations should also be raised and their capacity should be increased.

The urban's spatial structure may also influence commuting carbon emissions of residents' taxi travel. Our experiments show that carbon emissions from urban settlements in Xiasha Cheng, which is located far from the main urban area, show a low-low aggregation pattern of carbon emissions. In contrast, on the north side of Jiangnan Cheng closer to the central city area, there is an apparent high aggregation of carbon emissions from residential locations. The study area consists mainly of the central and two secondary cities, Xiasha Cheng and Jiangnan Cheng. The main function of Xiasha Cheng is to serve as an integrated new city with an economic and technological development zone and a high education park as its mainstay. Under the visionary plan, Jiangnan Cheng functions as a high-tech industrial and residential area and business core. A common feature of these three areas is a difference in land use.

Previous studies have shown that land-use patterns and land-bearing functions significantly affect carbon emissions via taxi travel. For example, Ref. [43] investigated the relationship between land properties, land use practices and positive and negative carbon transformations. Ref. [44] determined the relationship between land use types and "source–sink" patterns from taxi GPS data for Shanghai, and their results show a strong correlation between these variables. Ref. [45] measured land use types with 95% accuracy using the pick-up and drop-off point data of 4000 taxi trajectories for a year. Ref. [46], in a study of urban clusters using taxi GPS data, showed that different urban functions might lead to travel clusters of different sizes. Refs. [47,48]'s research shows that cities with a better mix of urban structures and road structures have lower carbon emissions when they expand, whereas cities with a single core have higher carbon emissions when they expand. These results suggest that urban functions and function mixes may influence the residential travel

carbon emissions of taxis and hence commuting carbon emissions. However, some studies have shown that excessive travel distance may lead to a preference for using taxis to travel with a threshold value of 15 km [49], which is similar to the distance from the central city area to Xiasha Cheng. From the results of previous experiments and studies, we speculate that this phenomenon may be due, on the one hand, to the fact that Xiasha Cheng is located far from the main urban area and is a fully functional integrated new city with better jobs, housing, and functions. On the other hand, this may be the case because the north side of Jiangnan Cheng is more homogeneous and closer to the main urban area, resulting in an imbalance of jobs and functions at the regional level. Therefore, in Hangzhou's future urban master plan, attention should be given to strengthening the balance of jobs and housing and to the coordination of urban functions in the Jiangnan Cheng area.

## 5. Conclusions

China's rapid urbanization has led to severe air pollution, and the transport sector has always accounted for a high proportion of $CO_2$ emissions. While representing a small proportion of transport carbon emissions, commuting carbon emissions have been growing at a relatively fast rate. At the same time, with the development of internet technology and big data technology, it is possible to conduct large-scale commuting surveys. As a new version of the urban master plan for Hangzhou is currently being prepared, the main purpose of this research is to provide a new way of studying commuting. The second purpose is to reveal the spatial and temporal characteristics of carbon emissions from commuting in Hangzhou and provide constructive suggestions for Hangzhou urban master planning to reduce commuting carbon emissions.

First, this study improves a method for measuring taxi commuting carbon emissions based on the VSP models with the G2SFCA method. We then use spatial autocorrelation analysis and Getis-Ord general Gi* regression analysis to explore the spatial and temporal characteristics of carbon emissions of residents' transport from taxi trips and the factors that may affect residents' taxi commuting carbon emissions. The study results indicate that carbon emissions in the central area of Hangzhou show a typical monocentric peak with a gradual decay towards the urban-rural border. The carbon emissions of taxis from 6:30–24:00 are 52,634 kg. Areas with high carbon emissions all day are mainly concentrated in the main urban area, with a small proportion concentrated on the northern side of Jiangnan Cheng and Xiasha Cheng. In the morning and evening peak hours, i.e., 6:30 a.m.–10:00 a.m. and 4:30 p.m.–8:00 p.m., taxi commuting carbon emissions reach 7228 kg and 2563 kg, respectively. Most people use taxis to commute from home to work.

We found that urban public transport facilities influences the commuting carbon emissions of residents. From the perspective of urban infrastructure development, residential locations with high-high aggregation of carbon emissions from taxi trips were evident near areas with poor public transport, especially in terms of metro density, such as in the urban-rural junction of the central city area. In contrast, areas with better public transport, such as the main urban area, appear as distinct settlements where carbon emissions from taxi travel are low-low aggregated. In terms of policy recommendations for urban planning at the level of urban presidential planning, we should strengthen the balance of jobs and housing and the construction of supporting jobs on the north side of Jiangnan Cheng. To strengthen the scale of population and urban spatial structure, the coordination of urban function distribution arrangement should be pursued to alleviate regional traffic pressure.

The coordinated arrangement of population size, urban spatial structure, and urban function distribution should be improved to ease traffic pressure in the region.

The public transit system in the city should be improved, and the density of public transit facilities in the city's fringe areas should be increased, such as the density of bus stops and railway stations. Public transport operation services must be enhanced and their capacity must be increased. A shift in urban transport from car-based transport to public transport will also be necessary to ease the pressure of carbon emissions from urban commuting.

**Author Contributions:** Conceptualization, C.X.; methodology, C.X. and S.L.; software, S.L. and J.Z. (Jun Zhou); formal analysis, S.L. and A.B.; resources, C.X. and J.Z. (Jun Zhou); data curation, S.L.; writing—original draft preparation, S.L. and Y.L.; writing—review and editing, C.X., F.X., S.L. and J.Z. (Jian Zhang); visualization, S.L. and A.B.; supervision, C.X., F.X. and J.Z. (Jian Zhang); project administration, C.X. and F.X.; funding acquisition, C.X. All authors have read and agreed to the published version of the manuscript.

**Funding:** This work was supported by the National Natural Science Foundation of China (Grant No. 72004014).

**Institutional Review Board Statement:** Not applicable.

**Informed Consent Statement:** Not applicable.

**Data Availability Statement:** (1) The taxi GPS data for this study was acquired from Comprehensive Information Centre of the Hangzhou Municipal Bureau of Transportation. (2) The POI data that we used comes from the Gao-De platform, crawled through the API interface of the Gaode developer platform: https://lbs.amap.com/ [50] accessed on 19 May 2022.

**Conflicts of Interest:** The authors declare that they have no conflict of interest.

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
