# Peer review of "A Big Data-Based Commuting Carbon Emissions Accounting Method—A Case of Hangzhou"

_land, doi:10.3390/land11060900_

Round 1

Reviewer 1 Report

Paper is interesting from climate change mitigations point of view, although figures are not clear for me.

Author Response

Dear reviewer:

Thank you very much for reviewing my paper in the midst of your busy schedule, and for the unclear diagram you mentioned. I have replaced all the figures in the latest version.

 Best regards

Reviewer 2 Report

This article focuses on current and important issues highlighting the relationship between air pollution and transport. Numerous studies have shown that air pollution causes a wide range of health consequences. They can be more or less life-threatening. One of the most troublesome manifestations of ambient air pollution for people and the environment is the accumulation of pollutants in the ground layer. There are several dominant sources of anthropogenic air pollution emissions. These are: combustion of fossil fuels for heating purposes in the municipal and household sector, in transport and electricity production and in industry. Other sources include waste treatment and agriculture. Natural sources include Volcanic eruptions and emissions of volatile organic compounds from plants. However, combustion processes outside industry, and thus mainly low emissions - burning coal in domestic cookers and boilers - are responsible for almost half of all PM10 emissions. Road transport has led to 9% of PM10 emissions (it is worth mentioning that it is not only fuel combustion, but also abrasion of brake pads, road surfaces and secondary dust). Dynamically growing car traffic as a source of air pollution is one of the most important development challenges of contemporary cities. The damage it causes generates real costs, which are increasingly calculated by economists. Even if the cost calculation is still imperfect, it makes it possible to realise how strong and negative the impact of combustion vehicles on the environment and our health is. However, despite the development of research raising awareness of this harmfulness, the scale of the problem is reaching the public very slowly.

The article has an adequate theoretical basis, relevant information and analysis, good partial (in the article) and final (in the conclusion) conclusions. The article uses original research by the author and cited research by other researchers, which enriches its content. It is written in good language and is based on the analysis of current and well-chosen literature, although it could be further enriched with other items. 

The research models were applied correctly. Systematics of models is not a simple issue, as the differentiation of model types results mainly from their precisely defined purpose. Therefore, the article should be treated as an interesting introduction to a very important issue and treated as a scientific article. The abstract of the article lacks information on the research method.

Author Response

Dear Reviewer:

         Thank you very much for your detailed and informative review of this article, which resulted in many constructive comments and ideas. I have made changes to all the issues you have mentioned.

  1. I have added some support from the literature that concentrates on transport carbon emissions differently. For example, low-carbon studies in the direction of new energy sources have been added to the introduction as support for the literature.
  1. In response to your comment about the lack of methods in the Abstract, I have added a description of the relevant methods in the latest version.

We look forward to hearing from you regarding our submission. We would be glad to respond to any further questions and comments that you may have.

 Best regards

Reviewer 3 Report

Dear authors

This paper presents the high accuracy method for estimating commuting carbon emissions in Hangzhou area using VSP model with a big data such as taxi GPS data.

It is very significant for urban planning to measure the accuracy data.

The process of the estimating commuting carbon emission was explained clearly.

However, there are some points which are needed to confirm and correct before publication.

Those are:

1)    In “Conclusion” section explained the methods which estimate commuting carbon emission in this research can effectively assist urban planning and achieve precise urban emissions reduction. I have the question, is it the possible for driver to suggest the best taxi route considering the low environmental burden? I guess the method including VSP model in this research for calculating the commuting carbon emission is able to contribute the low environmental burden timely consider the traffic condition timely.

2)    I think the figure number 1 and 2 are inverse. I suggest the change the number.

3)    In “Materials and Methods” section, figure 1 shows us the research process, however, no explanation of the figure 1. Please add the explanation.

I would like to confirm above the review and revise the manuscript.

Best regards

Author Response

Dear Reviewer:

Thank you very much for your careful reading and understanding of this study. It has pointed out many issues and shortcomings that our team had not considered. This is very important to improve our work and understanding in the future. For the issues, you have mentioned. I have changed each of them in the latest version.

  1. taxi drivers taking the best routes to reduce the environmental burden. Research has confirmed that good route planning can have a significant impact on reducing carbon emissions from travel. For example, the article: A study on the difference in carbon emissions between traditional taxis and online taxis in the city of ChengDu, China, showed that(https://doi.org/10.1016/j.jclepro.2019.04.159). The carbon emissions of traditional taxis are 1.4 times , etc., higher than the carbon emissions of online ride-hailing taxis. This article states. The study noted that online hailing taxis would wait for the platform to assign them after completing a task and then proceed directly to their destination, while traditional taxis would need to drive a cab around to search for task after completion. This is the reason for the difference in carbon emissions. This article is interesting. But the focus of this study is on the study of the relationship between urban structure and commuting carbon emissions; for driver path planning, I think it can be further improved in future studies.
  2. Figure 1 and Figure 2 are reversed for the second issue you mentioned. I have corrected this error. In the meantime, as Figure 1 is too complex, I have changed it to a more concise and understandable version.
  3. For the missing explanation of Figure 1 in the Materials and Methods section in Chapter 2. I have added a detailed explanation.

We look forward to hearing from you regarding our submission. We would be glad to respond to any further questions and comments that you may have

Best regards